# Non-feature-specific elevated responses and feature-specific backward replay in human brain induced by visual sequence exposure

Tao He[1,2,3,4], Xizi Gong[3,4], Qian Wang[3,4], Xinyi Zhu[3,4], Yunzhe Liu[5], Fang Fang[3,4,6,7]*

[1]Center for the Cognitive Science of Language, Beijing Language and Culture University, Beijing, China; [2]Key Laboratory of Language Cognitive Science (Ministry of Education), Beijing Language and Culture University, Beijing, China; [3]School of Psychological and Cognitive Sciences and Beijing Key Laboratory of Behavior and Mental Health, Peking University, Beijing, China; [4]IDG/McGovern Institute for Brain Research, Peking University, Beijing, China; [5]Chinese Institute for Brain Research, Beijing, China; [6]Peking-Tsinghua Center for Life Sciences, Peking University, Beijing, China; [7]Key Laboratory of Machine Perception (Ministry of Education), Peking University, Beijing, China

*For correspondence:
ffang@pku.edu.cn

## eLife Assessment

This **valuable** study investigates both online responses to, and offline replay of, visual motion sequences. Sophisticated MEG analyses provide **convincing** evidence for both feature-specific and non-specific sequence representations. These intriguing findings will be of interest to perception and learning researchers alike.

**Abstract** The ability of cortical circuits to adapt in response to experience is a fundamental property of the brain. After exposure to a moving dot sequence, flashing a dot as a cue at the starting point of the sequence can elicit successive elevated responses even in the absence of the sequence. These cue-triggered elevated responses have been shown to play a crucial role in predicting future events in dynamic environments. However, temporal sequences we are exposed to typically contain rich feature information. It remains unknown whether the elevated responses are feature-specific and, more crucially, how the brain organizes sequence information after exposure. To address these questions, participants were exposed to a predefined sequence of four motion directions for about 30 min, followed by the presentation of the start or end motion direction of the sequence as a cue. Surprisingly, we found that cue-triggered elevated responses were not specific to any motion direction. Interestingly, motion direction information was spontaneously reactivated, and the motion sequence was backward replayed in a time-compressed manner. These effects were observed even after brief exposure. Notably, no replay events were observed when the second or third motion direction of the sequence served as a cue. Further analyses revealed that activity in the medial temporal lobe (MTL) preceded the ripple power increase in visual cortex at the onset of replay, implying a coordinated relationship between the activities in the MTL and visual cortex. Together, these findings demonstrate that visual sequence exposure induces twofold brain plasticity that may simultaneously serve for different functional purposes. The non-feature-specific elevated responses may facilitate general processing of upcoming stimuli, whereas the feature-specific backward replay may underpin passive learning of visual sequences.

## Introduction

The capacity of cortical circuits to undergo plasticity in response to experience is a fundamental feature of the brain (*Buonomano and Merzenich, 1998*; *Costandi, 2016*; *Li, 2016*). This plasticity can be induced not only by active, task-dependent training, such as visual perceptual learning (*Watanabe and Sasaki, 2015*; *Chen et al., 2016*; *Lu and Dosher, 2022*), but also by passive, repetitive exposure (*Gutnisky et al., 2009*; *Sasaki et al., 2010*; *Ekman et al., 2017*). For instance, after repeated exposure to a moving dot sequence, flashing a dot (i.e. cue) at the starting point of the sequence elicits successive elevated neural responses in visual cortex, akin to those induced by the actual moving dot sequence in both humans (*Ekman et al., 2017*; *Ekman et al., 2023*; *Lu et al., 2021*) and animals (*Eagleman and Dragoi, 2012*; *Xu et al., 2012*). This cue-triggered reactivation is thought to be driven by expectations and is prediction-related (*Ekman et al., 2017*; *Ekman et al., 2023*), as it facilitates the prediction of upcoming stimuli and influences the perception of sensory information (*Gutnisky et al., 2009*; *Baker et al., 2014*; *Pojoga et al., 2020*).

Unlike the simple white dot sequences used in previous studies (*Xu et al., 2012*; *Ekman et al., 2017*; *Ekman et al., 2023*; *Lu et al., 2021*), temporal sequences to which we are exposed in daily life usually contain rich feature information. However, it remains unknown whether cue-triggered elevated responses are feature-specific. On the one hand, if these responses are not feature-specific, they may reflect a general state of cortical readiness for any upcoming stimuli. On the other hand, if these responses are indeed specific to a particular feature in the sequence, they would selectively facilitate the processing of specific future events. This aligns with the view that expectation sharpens neural tuning and enhances the processing of expected stimuli (*Kok et al., 2012*). For instance, expectation has been shown to elevate the pre-stimulus baseline activity of sensory neurons tuned to expected stimuli (*Wyart et al., 2012*; *Kok et al., 2014*) and to preactivate stimulus templates in both the visual (*Kok et al., 2017*) and auditory (*Demarchi et al., 2019*) cortices. Notably, however, all these prediction-related, feature-specific activities were observed in static contexts. Whether prediction-related responses in dynamic temporal contexts (e.g. visual sequences) are feature-specific remains unclear.

After exposure to a feature-contained temporal sequence, another important question is how the feature information in the sequence is encoded and organized in neural activity following the cue. Previous studies have demonstrated that memory consolidation (*Carr et al., 2011*; *Gridchyn et al., 2020*; *Gillespie et al., 2021*) or learning process *Igata et al., 2021*; *Liu et al., 2021b*; *Liu et al., 2022* following training with a temporal sequence is associated with a neural phenomenon known as replay. Replay refers to the sequential reactivation of neural activity patterns associated with the trained sequence in both sleep (*Lee and Wilson, 2002*) and awake (*Foster and Wilson, 2006*) states. During replay, the neural representation of the trained sequence is temporally compressed and can occur in either a forward or backward direction (*Carr et al., 2011*; *Joo and Frank, 2018*; *Nour et al., 2021*; *Liu et al., 2022*; *McFadyen et al., 2023*). These replay events frequently coincide with sharp-wave ripples (SWRs), which are high-frequency (150–220 Hz) oscillations (*O'Keefe and Nadel, 1978*; *Bush et al., 2022*) detected in hippocampal local field potentials (*Buzsáki, 1986*; *Buzsáki, 2015*). To date, replay has been observed in tasks involving sequences, such as nonlocal reinforcement learning (*Liu et al., 2021b*) and episodic memory retrieval (*Wimmer et al., 2020*). It remains unclear whether simple visual exposure to temporal sequences could also trigger replay events in humans.

In the current study, we used magnetoencephalography (MEG) to investigate whether cue-triggered elevated brain responses following visual sequence exposure are feature-specific and, more importantly, to determine how feature information in the exposed sequence is encoded and organized in the brain after exposure. To address these questions, participants were initially exposed to a predetermined motion sequence. We then decoded motion direction information during the blank period following the presentation of either the first or last motion direction in the sequence as a cue. Surprisingly, we found that cue-triggered elevated responses were not specific to any motion direction. Interestingly, motion direction information was spontaneously reactivated during the blank period, and the motion sequence was backward replayed in a time-compressed manner. This backward replay was identified even after brief exposure. However, neither forward nor backward replay was detected when an intermediate motion direction in the sequence was presented as a cue. Lastly, MEG source reconstruction analysis revealed that medial temporal lobe (MTL) activation preceded

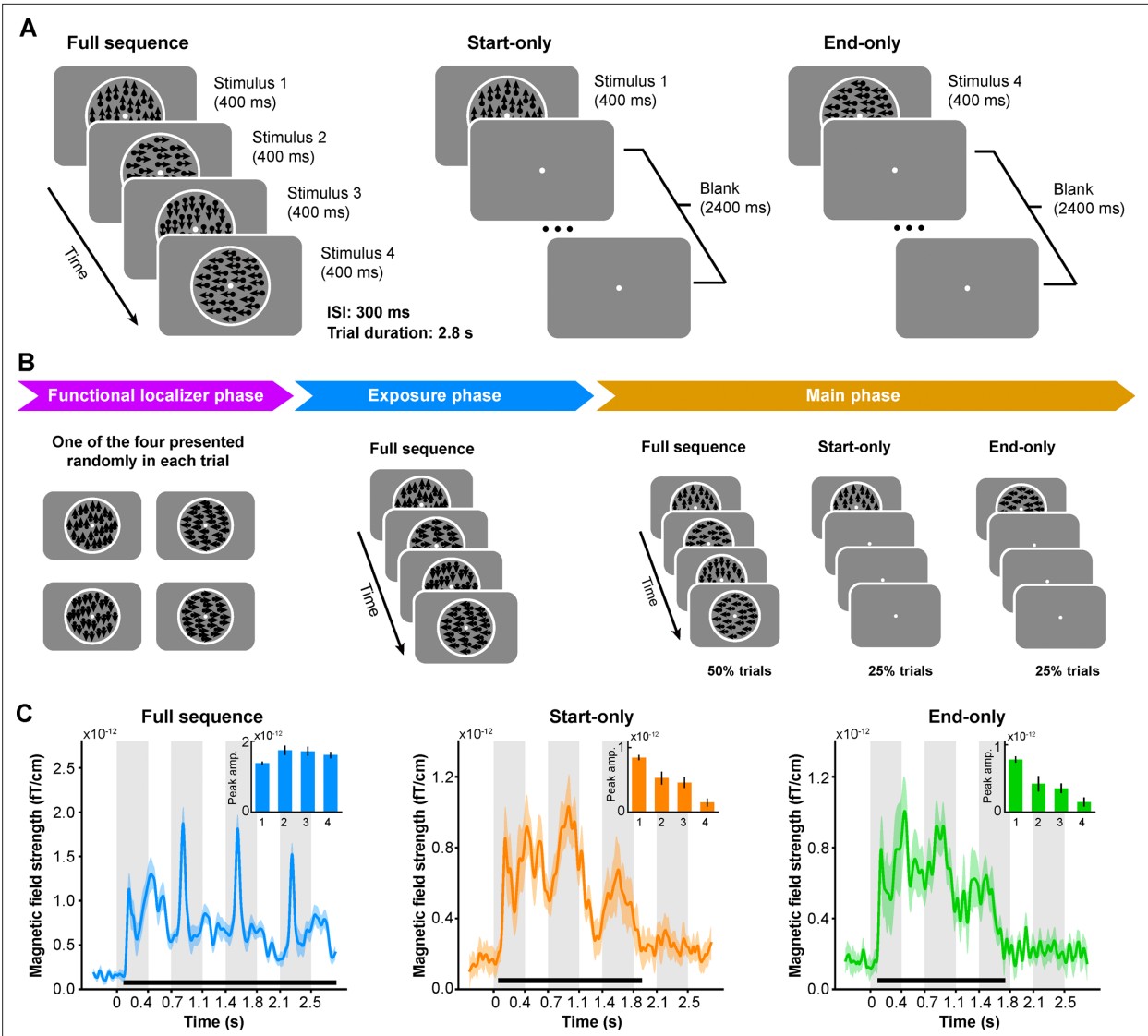

**Figure 1.** Stimuli, experimental procedure, and evoked responses in Experiment 1. (**A**) Participants were presented with random-dot kinematograms (RDKs) in three conditions. In the full sequence trial condition, four successive RDKs were presented. In the start- or end-only condition, only the first or last RDK in the sequence was presented at the beginning of the trial. (**B**) During magnetoencephalography (MEG) scanning, the participants were presented with two functional localizer runs (i.e. functional localizer phase) before providing any sequence information. Next, they were exposed to four full sequence runs (i.e. exposure phase). Finally, in the main phase, full sequence, start- and end-only trials were presented in a pseudorandomized order. (**C**) Evoked neural responses as a function of time relative to trial onset (n = 18) in the full sequence, start- and end-only trial conditions, respectively. Bold black lines at the bottom indicate temporal clusters in which they reached significance when compared to the pre-stimulus baseline. Inset figures at the top-right corner show the peak amplitudes during the four corresponding RDK intervals after baseline correction.

visual cortical activation, implying that the replay events observed in visual cortex may be triggered by activities in the MTL.

## Results

The visual stimuli used in the study were four random-dot kinematograms (RDKs). All dots in an RDK moved in a single direction (i.e. 0°, 90°, 180°, or 270°) with 100% coherence. In Experiment 1, we included three trial conditions: full sequence trials, start-only trials, and end-only trials (*Figure 1A*). In a full sequence trial, participants were exposed to a predefined sequence of the four RDKs presented either clockwise (i.e. 0° → 90° → 180° → 270°) or counterclockwise at the center of the screen, with an interstimulus interval of 300 ms between every two RDKs. In a start- or end-only trial, the first or

last RDK of the sequence was presented at the beginning of the trial, followed by a blank period of 2.4 s. Participants were instructed to complete three successive phases: functional localizer phase, exposure phase, and main phase (*Figure 1B*). The functional localizer data were used to train models to decode each motion direction in the sequence. During this phase, one of the four RDKs was randomly presented for 1 s in each trial. During the exposure phase, participants were exposed only to full sequence trials for about 30 min. In the main phase, 50%, 25%, and 25% of all trials were full sequence, start- and end-only trials, respectively. The full sequence trials served as a topping-up exposure to maintain the exposure effect.

## Cue-triggered elevated responses are not feature-specific

We first measured the event-related field (ERF) activity evoked by RDKs in the three trial conditions using all occipital gradiometer sensors (see Materials and methods). *Figure 1C* shows the evoked responses for full sequence trials (left panel, cluster-based permutation test with a cluster forming threshold of t>3.1 and 5000 permutations). Remarkably, start- and end-only trials elicited similar wave-like responses as the full sequence trials, despite the absence of stimuli following the first RDK (*Figure 1C*, middle and right panels, cluster-based permutation test with a cluster forming threshold of t>3.1 and 5000 permutations). To further quantify the ERF peak amplitudes in the three conditions while mitigating baseline confounds, we calculated each peak amplitude using the 300ms blank period just preceding the onset of the corresponding RDK as the baseline (see Materials and methods). We found that the four successive RDKs in the full sequence trials evoked four comparable peaks after stimulus onset (*Figure 1C*, inset figure in the left panel; two-tailed t-test; all $ts_{(17)} > 6.8072$, all ps<$10^{-5}$). Interestingly, in start- and end-only trials, significant peaks were still observed during the periods corresponding to the intervals of the second and third RDKs in the full sequence trials (*Figure 1C*, inset figures in the middle and right panels; start-only condition, two-tailed t-test; all $ts_{(17)} > 4.5790$, all ps<$10^{-3}$; end-only condition, two-tailed t-test; all $ts_{(17)} > 6.0140$, all ps <$10^{-4}$). However, no significant peak was observed during the period corresponding to the interval of the last RDK (start-only condition, two-tailed t-test; $t_{(17)} = 1.3221$, p=0.2036; end-only condition, two-tailed t-test; $t_{(17)} = 0.2473$, p=0.8076). These results demonstrate cue-triggered elevated neural responses following visual exposure to the RDK sequences, resembling previous studies using simple white dot sequences (*Ekman et al., 2017*; *Lu et al., 2021*).

Given that we observed elevated responses even in the absence of stimuli following the cue, we next examined whether these responses were specific to a particular feature (i.e. motion direction). Specifically, we asked whether motion directions could be successfully decoded in start- and end-only trials, particularly during the blank periods corresponding to the intervals of the second and third RDKs in full sequence trials. To this end, we applied a time-resolved decoding analysis. For each participant, we trained a one-versus-rest Lasso logistic regression model using the functional localizer data to classify the neural activity pattern elicited by each motion direction in the main phase. MEG signals from 72 occipital sensors were selected as features for the training model.

To validate the reliability of our model, we first used a leave-one-out cross-validation scheme on the localizer data to independently estimate decoding accuracy at each time point. At the group level, decoding accuracies peaked at 411 ms, 464ms, 444 ms, and 434 ms after stimulus onset for motion directions 0° (55.37%±1.21), 90° (58.10%±1.14), 180° (54.07%±1.47), and 270° (53.64%±1.34), respectively. There were no significant differences among the four motion directions in terms of either latency ($F(3, 51)=0.375$, p=0.7716, $\eta_p^2=0.0159$) or decoding accuracy ($F(3, 51)=2.757$, p=0.0517, $\eta_p^2=0.091$) at the peak time point. Next, we trained a model using the localizer data averaged between 100 ms and 500 ms after stimulus onset. The trained model was then applied to the MEG signals recorded in the three trial conditions in the main phase. Finally, we calculated the decoding probability (*Liu et al., 2019*; *Nour et al., 2021*; *Turner et al., 2023*) for each motion direction at each time point at the group level (*Figure 2*). Here, decoding probability for each motion direction reflects the likelihood that the decoded stimulus had a specific motion direction at a given time point (see Materials and methods). Therefore, it provides a time-resolved decoding preference for each motion direction, rather than only a single decoded label (e.g. 0°, 90°, 180°, or 270°).

In full sequence trials, the motion direction information could be successfully decoded, as evidenced by the highest decoding probability during the interval corresponding to the presentation of the respective RDK (*Figure 2A and B*, left panels). In start- and end-only trials, we could also

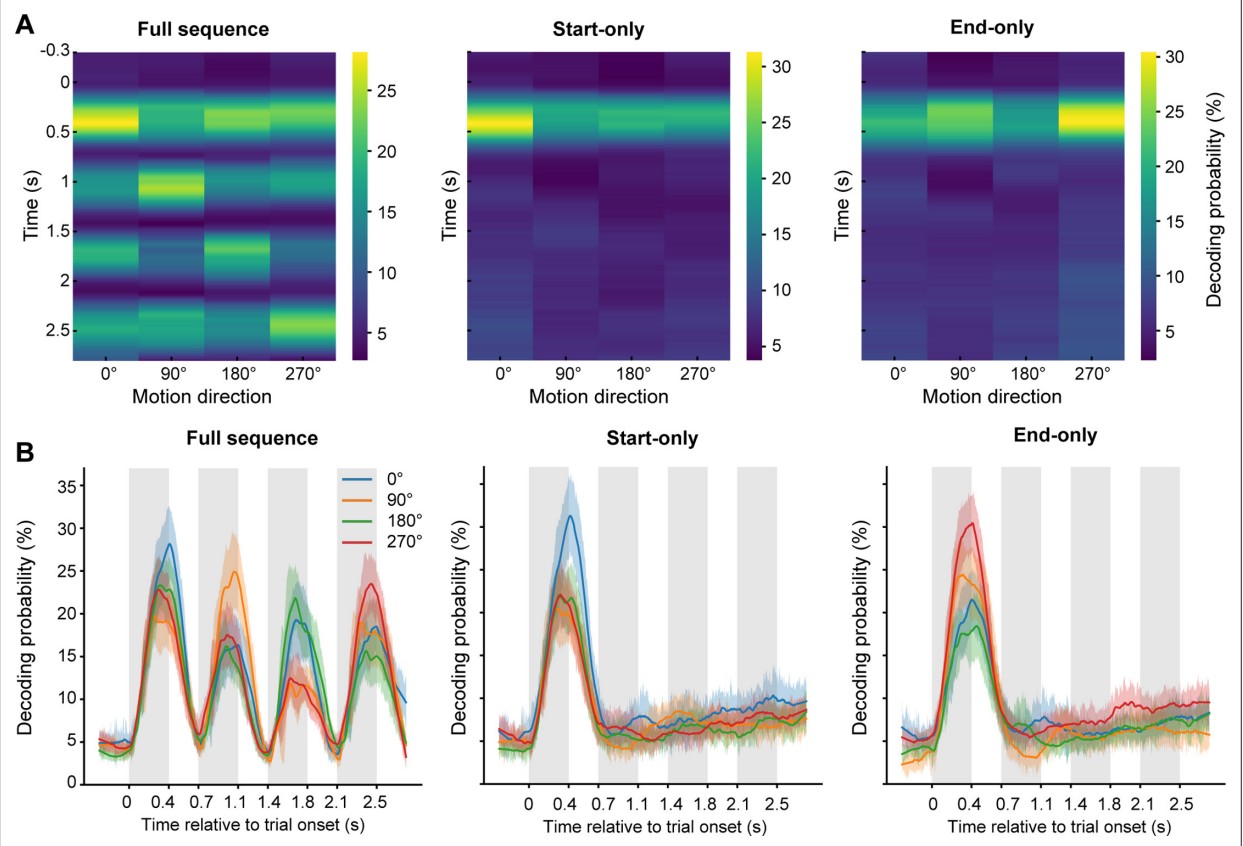

**Figure 2.** Time-resolved decoding probability for each motion direction. (**A**) Visualization of the time-resolved decoding probability in the three trial conditions. Each row shows the decoding probabilities for the four motion directions at that time point, and each column indicates one of the four motion directions. (**B**) The line plots of the time-resolved decoding probability for the three trial conditions. Each colored line shows the time course of the decoding probability for each motion direction. For the start- and end-only conditions, we calculated the permutation threshold estimated by randomly shuffling the labels and re-decoding; only the decoding probability of the cue surpassed the threshold.

reliably decode the motion direction of the first RDK (i.e. the cue) after stimulus onset (*Figure 2A and B*, middle and right panels, only the decoding probability of the first RDK surpasses the peak-level significance threshold obtained from a nonparametric permutation test, FWE corrected across time). Surprisingly, however, subsequent motion direction information was absent at the group level during the post-cue blank period, where the cue-triggered elevated responses were previously observed (i.e. 0.4–2.8 s after stimulus onset). Together, these results demonstrate that the cue-triggered elevated response induced by visual sequence exposure was not consistently specific to a particular feature across participants and trials.

## Time-compressed backward replay of exposed motion sequence

How is the motion direction information encoded and organized in the brain during the post-cue blank period? Clearly, the motion direction representation is not time-locked to the onset of the cue in either start- or end-only trials. However, it is possible that individual motion directions are encoded in the MEG signals in a spontaneous way but are sequentially organized (*Liu et al., 2019*).

To test this hypothesis, we trained four decoding models using the localizer data to capture the neural features of the four motion directions. We employed four decoding models because our aim was to build feature-specific classifiers, each sensitive to only one motion direction. These classifiers were designed to quantify the evidence of feature-specific sequence in subsequent analyses. The models used a one-versus-rest Lasso logistic regression algorithm, with MEG signals from 72 occipital sensors as features (*Figure 3A*). For each participant and motion direction, we then selected the time point with the highest decoding accuracy, which was estimated during the validation of the functional localizer data, as the optimal time point. These optimal time points were chosen because they are

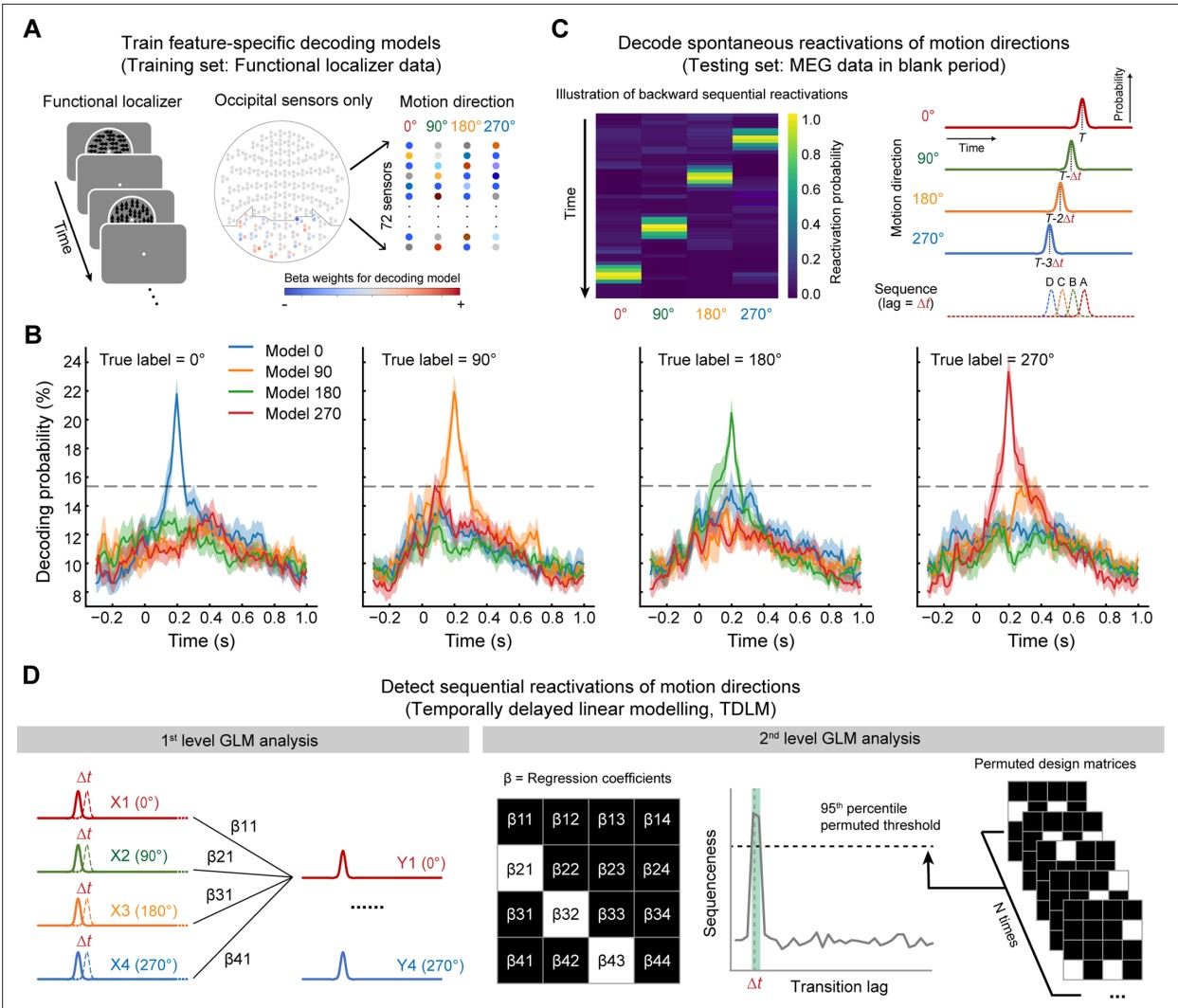

**Figure 3.** Illustration of replay analysis pipeline: temporally delayed linear modeling (TDLM). (**A**) A Lasso logistic regression model was trained for each participant and motion direction using magnetoencephalography (MEG) signals from the functional localizer data. (**B**) The four feature-specific models that were trained using functional localizer data. The decoding probabilities were aligned at 200 ms poststimulus onset according to their corresponding optimal time point per participant and motion direction. Each colored line indicates the decoding results of a model applied to a motion direction dataset. The dashed horizontal line indicates the permutation threshold estimated by random shuffling of the labels and re-decoding. (**C**) The four models were next applied to MEG signals during the post-cue blank period to derive a decoded reactivation matrix [time × motion direction]. An illustration of backward sequential reactivations of motion direction is shown on the left. Reactivation probabilities correspond to the decoding probabilities derived from the four models. (**D**) Using TDLM, we quantified the evidence for sequential replay of the motion sequence during the post-cue blank period in start- and end-only conditions. We first performed a time-lagged regression to generate a [4×4] empirical regression coefficient matrix for each time lag by regressing each lagged predictor matrix, X(Δt), onto the original reactivation matrix, Y (i.e. first-level GLM analysis). Next, we used a second-level GLM analysis to evaluate the extent to which the empirical transition matrix follows a model transition matrix (e.g. forward or backward transitions) (i.e. second-level GLM analysis). Finally, we calculated the difference between the second-level regression coefficients for forward and backward transitions, referred to as 'sequenceness'. We tested the magnitude of this 'sequenceness' at each time lag independently for all transition lags up to 600 ms. The dashed line represents the corrected nonparametric statistical significance threshold. The green area indicates the lags when the evidence of 'sequenceness' in the backward direction exceeded the permutation threshold.

The online version of this article includes the following figure supplement(s) for figure 3:

**Figure supplement 1.** Sensor maps and spatial correlation of trained Lasso logistic regression models.

**Figure supplement 2.** Decoded feature representations in start- and end-only conditions.

believed to carry the richest feature information (*Mo et al., 2019*). For subsequent analyses, we obtained four classifiers that were trained on the localizer data at their respective optimal time points. Each classifier yielded significant decoding probability only when the stimulus matched the motion direction it was trained to detect (*Figure 3B*). We did not find any spatial correlation between any two trained classifiers (highest correlation r<0.12, *Figure 3—figure supplement 1*). Since the optimal time point varied across participants and motion directions, we circularly shifted the decoding probabilities over time and aligned them to a common time point (arbitrarily set to 200 ms after stimulus onset) for visualization at the group level (*Figure 3B*).

Having established the classifiers for each participant and motion direction, we then applied the classifiers to the MEG signals during the post-cue blank period in start- and end-only conditions to estimate the reactivation probability (i.e. decoding probability) for each motion direction at each time point (*Figure 3C*; *Liu et al., 2019*; *Wimmer et al., 2020*; *Nour et al., 2021*). Example trials for the start- and end-only conditions are shown in *Figure 3—figure supplement 2*. The results revealed that motion direction reactivations occurred sparsely during the post-cue blank period, rather than crowded within intervals corresponding to the RDK presentations in the full sequence condition, suggesting that motion direction information might be reactivated in a spontaneous way. Next, we applied temporally delayed linear modeling (TDLM) to quantify whether and how these spontaneous reactivations followed the order of the exposed sequence (*Figure 3D*; *Kurth-Nelson et al., 2016*; *Liu et al., 2019*; *Nour et al., 2021*). This algorithm includes two-level regression analyses. The first-level

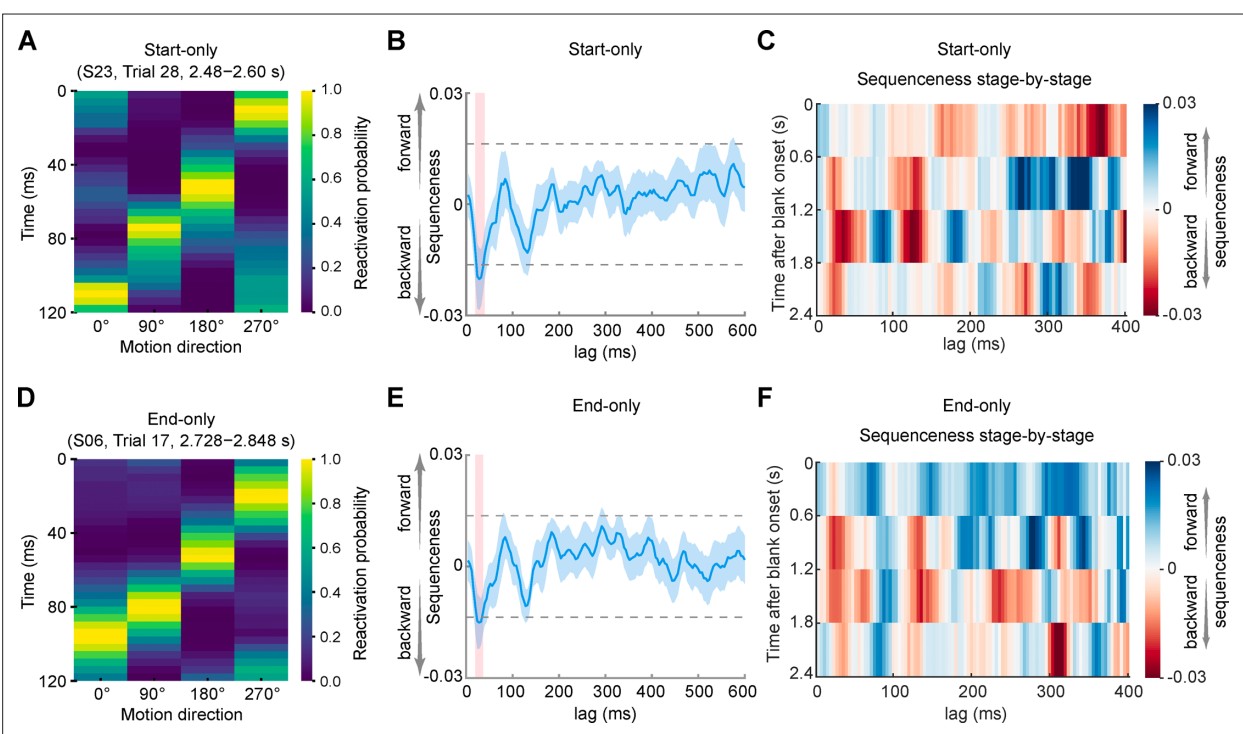

**Figure 4.** Backward replay in both start- and end-only conditions. (**A** and **D**) Examples of backward sequential reactivation in start- (**A**) and end- (**D**) only conditions from a representative participant. Each row represents the reactivation probabilities for the four motion directions at that time point, and each column indicates one of the four motion directions. (**B** and **E**) Backward replay of the exposed motion sequence with peak transition lags at 28–40 ms in the start-only condition (**B**) and at 28–36 ms in the end-only condition (**E**). Horizontal dashed lines represent corrected significance levels from a nonparametric permutation test at the second-level GLM analysis of temporally delayed linear modeling (TDLM). The sequenceness on the y-axis is a unitless measure. The lag on the x-axis represents the time lag (Δt) between sequential reactivations, rather than absolute time. The red shaded areas indicate the lags when the evidence of sequenceness exceeded the permutation-based significance threshold. (**C** and **F**) Backward replay of the exposed motion sequence predominantly appeared at 1.2–1.8 s in the start-only condition (**C**) and 0.6–1.8 s in the end-only condition (**F**) after the onset of the blank period.

The online version of this article includes the following figure supplement(s) for figure 4:

**Figure supplement 1.** Sequenceness distribution across participants.

**Figure supplement 2.** Sequenceness for each of the 24 possible orders.

regression quantifies the evidence for each pairwise transition (e.g. 0° → 90°), resulting in an empirical transition matrix. The second-level regression evaluates the extent to which this empirical transition matrix aligns with a specific sequence of interest. Finally, we defined 'sequenceness' as a metric of forward (i.e. 0° → 90° → 180° → 270°) or backward (i.e. 270° → 180° → 90° → 0°) replay (see Materials and methods).

As shown in *Figure 4A and B*, in the start-only condition, we found evidence of backward replay of the exposed motion sequence (i.e. the replay sequence was 270° → 180° → 90° → 0° when the exposed motion sequence was 0° → 90° → 180° → 270°) during the post-cue blank period, with a peak transition lag at 28–40 ms (maximal effect at 32 ms-lag: β = −0.0202±0.002, p<1/24 ≈ 0.042, peak-level significance threshold derived from a nonparametric permutation test, FWE corrected across lags, *Figure 4B*). For visualization, an example of backward replay of the motion sequence is illustrated in *Figure 4A*. This effect was observed in most participants (*Figure 4—figure supplement 1*). In the end-only condition, we also found evidence of backward replay during the post-cue blank period, with a peak transition lag at 28–36 ms (maximal effect at 32 ms-lag: β = −0.0145±0.0016, p<1/24 ≈ 0.042, peak-level significance threshold obtained from a nonparametric permutation test, FWE corrected across lags; *Figure 4E*). *Figure 4D* shows an example of backward replay of the motion sequence in this condition found in most participants (*Figure 4—figure supplement 1*). Note that the time lag in the horizontal axis in *Figure 4B and E* indicates the interval between the onsets of every two items in the replayed motion sequence. We found that the replayed sequence was approximately 10 times faster than the evoked activity sequence in the full sequence condition.

To further examine the period during which the backward replay occurred most frequently, we divided the blank period (2.4 s) into four stages, each lasting 600 ms. In start-only trials, we found that the backward replay predominantly appeared within the third stage of the blank period (1.2–1.8 s after the onset of the blank period, Wilcoxon signed-rank test, p=0.0108), but there were no significant sequenceness in the other three stages (first stage, p=0.7112; second stage, p=0.5566; fourth stage, p=0.6791; *Figure 4C*). In end-only trials, the backward replay was more likely to occur during the second and third stages of the blank period, although the results approached, but did not reach, significance (Wilcoxon signed-rank test, second stage, p=0.0936; third stage, p=0.0642; *Figure 4F*). In contrast, the replay was not frequently observed during the first and last stages (Wilcoxon signed-rank test, first stage, p=0.9479; fourth stage, p=0.4997). Finally, in a control analysis, we conducted a comprehensive examination of all 24 potential sequences. Only the backward replay was detected in both start- and end-only conditions (*Figure 4—figure supplement 2*).

## Backward replay is cue-dependent and depends on the amount of exposure

So far, we have demonstrated that the cue-triggered elevated responses induced by the motion sequence exposure were not motion direction specific. However, motion information was spontaneously reactivated, and the motion sequence was backward replayed in a time-compressed manner. It remains unknown whether the observed backward replay of the motion sequence was cue-dependent. In other words, does the replay of the motion sequence occur irrespective of which item of the sequence is presented as a cue? To address this question, we conducted Experiment 2, mirroring the design of Experiment 1 but with a different cue. Instead of flashing the first or last RDK, we presented the second or third RDK as a cue (second-only or third-only condition, see *Figure 5—figure supplement 1* and Materials and methods) to examine whether these two non-terminal cues could induce replay during the post-cue blank period. We found no evidence of either forward or backward replay in either condition (*Figure 5A*; maximal nonsignificant effect at 32 ms-lag: second-only condition, β = −0.0024±0.0014; third-only condition, β = −0.0018±0.0015).

Another interesting question is how varying the amount of exposure would affect replay events during the blank period. To explore this issue, we conducted Experiment 3, also similar to Experiment 1 except for the removal of the exposure phase. Accordingly, only full sequence trials (50% trials) in the main phase served for exposure (see *Figure 5—figure supplement 1* and Materials and methods). In the start-only condition, we found a numerical trend of backward replay at transition lags at 20–40 ms, although it did not reach statistical significance (*Figure 5B*, left panel; maximum effect at 28 ms-lag: β = −0.0139±0.0021). In the end-only condition, the evidence for backward replay just reached the significance level at transition lags at 20–40 ms (*Figure 5B*, right panel; maximal effect at

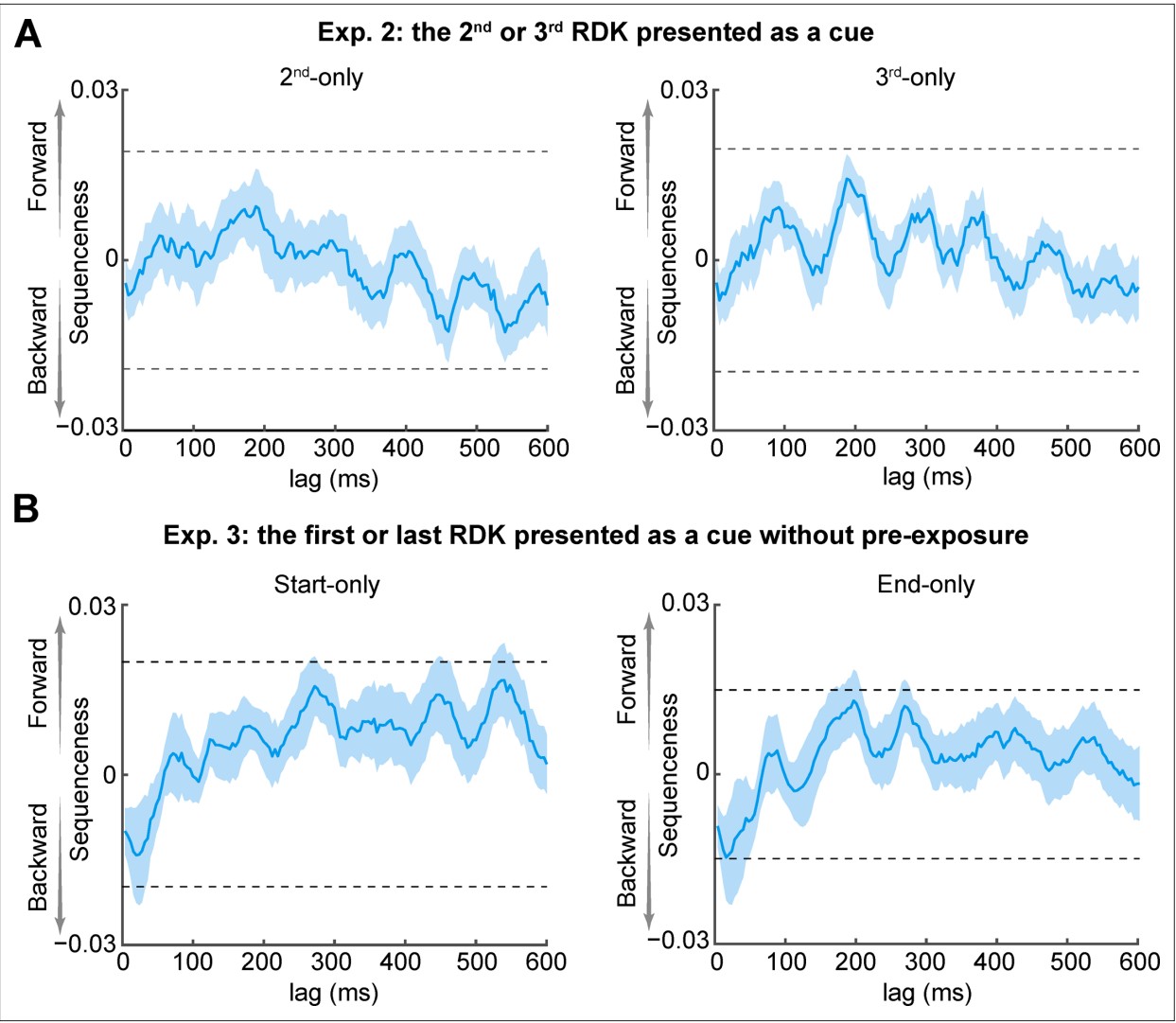

**Figure 5.** Backward replay is cue-dependent and depends on the amount of exposure. (**A**) No evidence for replay was found in either the second-only (left) or third-only (right) condition in Experiment 2. Horizontal dashed lines have the same meaning as those in *Figure 4B and E*. (**B**) In Experiment 3, immediately after the functional localizer phase, participants entered the main phase without the exposure phase. In the end-only condition, backward replay was observed; however, in the start-only condition, no such replay was observed.

The online version of this article includes the following figure supplement(s) for figure 5:

**Figure supplement 1.** Stimuli and experimental procedures in Experiments 2 and 3.

24 ms-lag: β = −0.0147±0.0019, p<1/24 ≈ 0.042, using the peak-level significance threshold from a nonparametric permutation test, FWE corrected across lags).

## Power increase in replay-associated SWR frequencies

Previous studies on rodents and humans showed that replay events are associated with increased high-frequency ripple power (*Buzsáki, 2015*; *Liu et al., 2019*). To investigate whether such replay-associated power increase could be observed in our study, we performed a time-frequency analysis using combined data from the start- and end-only conditions in Experiment 1. We first identified putative replay onsets during the post-cue blank period in each trial. Replay onsets were determined by choosing time points with a high (>95th percentile) probability for backward replay with a 32 ms-lag transition (maximal replay effect at the group level for both start- and end-only conditions, *Figure 4B and E*; see Materials and methods). Using the MEG signals recorded from whole-brain sensors, we found a transient ripple power increase at 120–180 Hz at the onset of replay events, compared to the

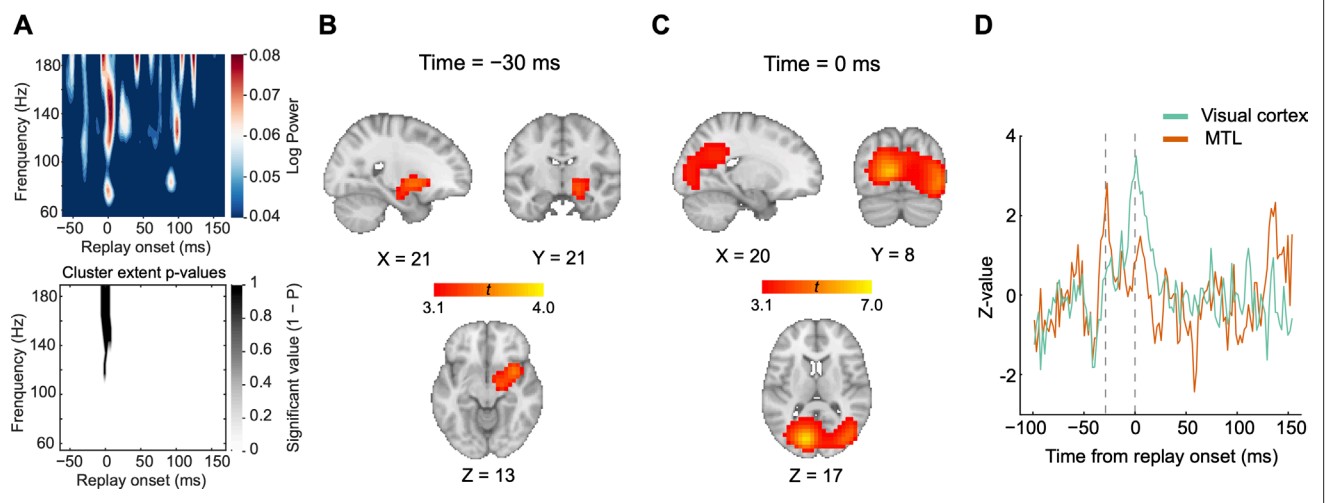

**Figure 6.** Replay events align with ripple power, with source activation in the medial temporal lobe (MTL) preceding activation in visual cortex. (**A**) Top: In Experiment 1, a time-frequency decomposition of sensor-level data revealing a brief increase in high-frequency oscillatory power at replay onset. Bottom: A cluster-based permutation test (cluster forming threshold, t>3.1; number of permutations = 5000) could identify a significant cluster around 140 Hz (n = 18). (**B**) Source localization of ripple-band power 30 ms before replay onset showing significant activation in the MTL (peak Montreal Neurological Institute [MNI] coordinates: X=21, Y=21, Z=13, neurological orientation). (**C**) Source localization of ripple-band power at replay onset showing significant activation in visual cortex (peak MNI coordinates: X=20, Y=8, Z=17). (**D**) The activation time course of the MTL at its peak MNI coordinate is shown in red, whereas that of visual cortex at its peak MNI coordinate is displayed in green. The MTL reached its peak activation before visual cortex.

baseline period of 50–100 ms prior to replay onset (*Figure 6A*; cluster-based permutation test with cluster forming threshold t>3.1 and 5000 permutations).

## Visual cortical activation lags MTL activation

Since replay events were obtained based on decoding probabilities of motion directions from occipital sensors only, we next examined whether the observed replay onsets were related to power increase within visual cortex. Using a linearly constrained minimum variance (LCMV) beamforming algorithm (*Van Veen et al., 1997*), we first epoched the data using replay onsets combined from both start- and end-only conditions and then beamformed the broadband power into the source space (see Materials and methods). We found that the power increase at replay onset was localized to sources in bilateral visual cortex (*Figure 6C*). Moreover, the right MTL, including the hippocampus and amygdala, was associated with power increase prior to the onset of replay events (*Figure 6B*). Specifically, activation (i.e. power increase) in the MTL occurred 30 ms earlier than that in visual cortex (p<0.05, corrected, whole-brain nonparametric permutation test). For display purposes, we extracted activations from the 10 most activated voxels within the MTL and visual cortex, respectively, and plotted the time courses of their broadband power. Peak activation in visual cortex at replay onset was preceded by peak activation in the MTL (*Figure 6D*), implying an information flow from the MTL to visual cortex.

## Discussion

We found the coexistence of time-locked, non-feature-specific elevated responses and non-time-locked, feature-specific backward replay after visual sequence exposure in human brain. The feature-specific backward replay occurred in a time-compressed manner and could be triggered only by the first or last stimulus of the sequence. Interestingly, even brief exposure to the sequence could still induce a trend for the backward replay. Finally, we observed that MTL activity preceded the ripple power increase in visual cortex at replay onset.

Our study provides two new findings in the fields of vision and learning. First, different from many previous studies showing that expectation-based responses in static contexts are feature-specific (*Wyart et al., 2012*; *Kok et al., 2014*; *Kok et al., 2017*), we show here that prediction-related elevated responses in the dynamic temporal context are not. This non-feature-specific elevated

response potentially facilitates the general processing of any upcoming stimuli, rather than stimuli with a specific feature. Second, contrary to the prevailing notion that replay in learning and memory requires lengthy training with a task, our study shows that even brief exposure to a visual sequence is sufficient to induce replay. The replay we unveiled here may play a critical role in memorizing the visual sequence.

Regarding the twofold neural consequences following the visual sequence exposure, the non-feature-specific elevated responses could be induced by rhythmic entrainment. Substantial evidence supports that rhythmic stimulation can entrain neural oscillations, which in turn facilitates predictions about future inputs and enhances the brain's readiness for incoming stimuli (*Lakatos et al., 2008*; *Lakatos et al., 2013*; *Herrmann et al., 2016*; *Barne et al., 2022*). For instance, recent findings demonstrate that even task-irrelevant information could be more effectively decoded when presented at moments close to a highly probable target presentation (*Auksztulewicz et al., 2019*). In our study, the rhythmic presentation of the motion sequence may have entrained oscillatory activity in the brain, leading to periodic activation of sensory cortices. This rhythmic entrainment likely serves as a possible mechanism supporting the general facilitation of neural processing for any upcoming stimuli, independent of specific stimulus features.

In contrast, feature-specific replay may operate through a different mechanism driven by intrinsic neural processes rather than direct external stimuli. Therefore, this intrinsic activity is not time-locked to stimulus onset, but instead manifests in a spontaneous way. Furthermore, since replay often occurs during offline periods (e.g. rest or blank period), it allows the neural system to reactivate and consolidate past experiences without interference from ongoing external inputs. As a result, feature-specific replay may facilitate the integration of fragmented experiences into coherent representations, thereby subserving visual sequence learning and memory.

Another interesting question regarding the twofold neural responses is whether the elevated responses and the backward replay share the same neural origin, for instance, originating from the hippocampus or other brain areas. For the elevated responses, the cue serves to provide temporal information for upcoming stimuli but without detailed feature information, thereby priming the cortices for activation and facilitating the general processing of future events. Previous animal studies (*Xu et al., 2012*; *Gavornik and Bear, 2014*) showed that this process can be implemented through a simple local synaptic mechanism in visual cortex (*Xu et al., 2012*) without top-down guidance. Moreover, *Ekman et al., 2023*, recently found no functional relationship between activities in V1 and the hippocampus when exposing human participants to a white dot sequence, further suggesting that the elevated responses may indeed originate in visual cortex.

Different from the elevated responses, the replay events are likely initiated by the hippocampus. This speculation is supported by two key findings. First, the replay of the motion direction information manifested in a backward direction and a time-compressed manner. Second, the replay is observed regardless of whether the cue is the first or last RDK in the sequence. Both of these two properties cannot be explained by a simple local synaptic mechanism (*Xu et al., 2012*) or a pattern completion-like mechanism (*Hindy et al., 2016*; *Ekman et al., 2017*; *Kok and Turk-Browne, 2018*) within visual cortex alone. Instead, the replay entails reorganizing the motion direction sequence in the brain. Therefore, we propose the involvement of active communication between the hippocampus and visual cortex in the occurrence of replay events, consistent with the view that the hippocampus encodes relationships among stimuli, whereas visual cortex primarily acts as a platform for the manifestation of replay events (*Whittington et al., 2020*). Previous studies on memory consolidation considered the exchange of information between these two regions critical for facilitating replay events through hippocampal-neocortical circuits (*Buzsáki, 1996*; *Ji and Wilson, 2007*; *Carr et al., 2011*; *Ólafsdóttir et al., 2016*; *Buch et al., 2021*). Functionally, replay events offer a mechanism for transferring recent experience from the hippocampus to the cortex, enabling the encoding of stimulus relationships in the cortex (*Marr and Brindley, 1971*; *Alvarez and Squire, 1994*; *Redish and Touretzky, 1998*; *Dimakopoulos et al., 2022*).

A natural behavioral consequence of visual sequence exposure is visual sequence learning (*Baker et al., 2014*; *Finnie et al., 2021*; *Ekman et al., 2023*). How is the learning implemented through hippocampus-dependent replay in visual cortex? Three key processes are considered here. First, the hippocampus is involved in encoding relationships among stimuli (*Staresina and Davachi, 2009*; *Turk-Browne et al., 2009*; *Hsieh et al., 2014*; *Garvert et al., 2017*); in fact, the ability to encode

such relationships drastically decreases when the hippocampus is damaged (*Chun and Phelps, 1999*; *Hannula et al., 2006*; *Konkel et al., 2008*; *Schapiro et al., 2014*; *Finnie et al., 2021*). Second, visual cortical areas act as a 'cognitive blackboard' where task-relevant features are highlighted through feedback connections (*Roelfsema and de Lange, 2016*). Such a blackboard can be flexibly written or edited. Third, there are strong bidirectional connections between the hippocampus and sensory cortices (*Eichenbaum et al., 2007*; *Henke, 2010*). As proposed by the hippocampal-cortical backward projection model (*Rolls, 2000*), sequential reactivations of feature information initially generated in the hippocampus can be quickly and accurately sent back to the sensory cortices, consistent with the findings of *Ji and Wilson, 2007*. A recent study also provided direct evidence that visual sequence plasticity is impaired when the hippocampus is damaged (*Finnie et al., 2021*), supporting the hypothesis of functional feedback information flow.

Why does the replay manifest in a reverse order? To date, there is no consensus on this issue. In rodents, a seminal study of replay during the awake state showed that when animals stopped at the end of a rewarded pathway, place cells were reactivated in the reverse order of the previously experienced direction (*Foster and Wilson, 2006*). However, a later study revealed that awake replay could occur in either a forward or backward direction relative to behavioral experience (*Diba and Buzsáki, 2007*). Similarly, in humans, both directions have been observed in different nonspatial cognitive tasks (*Liu et al., 2019*; *Wimmer et al., 2020*). Forward replay may be associated with planning (*Ólafsdóttir et al., 2015*; *Gillespie et al., 2021*), providing information pertaining to the assessment of future pathways (*Diba and Buzsáki, 2007*). Backward replay may be more related to experience, as often observed at the end of runs when animals consume a reward (*Foster and Wilson, 2006*; *Ambrose et al., 2016*). Nevertheless, the exact function of the replay direction remains mysterious, as both forward and backward replays are modulated by task demands (*Ólafsdóttir et al., 2017*). Thus, the underlying neural mechanisms of backward replay in visual cortex remain to be investigated.

Finally, we also found that replay occurrence is modulated by the cue. This result highlights the importance of the start and end points of the sequence in the replay. One fascinating proposal is that the replay event is sensitive to sequence boundaries, as indicated by the role of the start or end point of the sequence as salient boundaries that anchor place cell firing (*Rivard et al., 2004*). Accordingly, previous studies have shown that when rats are trained to run along a linear track starting at different points, place fields tend to be anchored to either the start or end of the journey (*Gothard et al., 1996*; *Redish et al., 2000*), suggesting that a boundary is essential to sequence integrity and may play a pivotal role in triggering replay onset. An alternative explanation to these findings posits that the onset of the second or third stimulus in the sequence reinstates the neural representations of a partial visual sequence (*Ekman et al., 2023*). For example, for a given sequence, A → B → C → D, flashing the third stimulus (i.e. C) only trigger a backward reactivation of the sequence giving C → B → A during the blank period.

Taken together, we found that simple visual sequence exposure could concurrently induce twofold brain plasticity, i.e., non-feature-specific elevated responses and feature-specific backward replay in the human visual cortex. We speculate that the non-feature-specific elevated responses may enhance general processing of upcoming visual stimuli, whereas the feature-specific backward replay may subserve visual sequence learning and memory. These findings significantly advance our understanding of the task independence and the multifaceted nature of brain plasticity in response to visual experience.

## Materials and methods

### Participants

A total of 59 healthy participants (29 females) were involved in the three experiments (Experiment 1: n=21, 11 females, 22.1±2.61 years; Experiment 2: n=18, 10 females, 23.56±3.29 years; Experiment 3: n=20, 10 females, 20.65±2.62 years). In Experiment 1, data from three participants were excluded before analyses, as two showed large head motion (>20 mm), and the behavioral performance of one was at chance level. In Experiment 3, data from two participants were excluded before analyses because of large head motion (>20 mm). No statistical methods were used to predetermine sample sizes, but our sample sizes are comparable to previous studies (*Liu et al., 2019*; *Mo et al., 2019*). All participants were recruited in exchange for monetary compensation (100 RMB/hr). They reported

normal or corrected-to-normal vision and had no history of neurological disorders. They were naive to the purposes of the study. The experiments reported here were carried out in accordance with the guidelines expressed in the Declaration of Helsinki. All participants provided written informed consent in accordance with the procedures and protocols approved by the Human Subject Review Committee of Peking University.

## Task

### Experiment 1

Visual stimuli were RDKs with 100% coherence. All dots in an RDK moved in the same direction (luminance: 2.86 cd/m$^2$; diameter: 0.1°; speed: 8°/s) and were presented against a gray background (luminance: 16.7 cd/m$^2$). At any one moment, 400 dots were visible within a 9° circular aperture and moved in one of the four directions: 0° (right), 90° (up), 180° (left), and 270° (down). Each participant completed three phases successively in the MEG scanner, namely, the functional localizer phase, the exposure phase, and the main phase (*Figure 1C*). In the functional localizer phase, each trial started with the presentation of an RDK for 1 s followed by a 1–1.5 s intertrial interval (ITI). Participants did not need to perform any task in this phase. The motion direction of the RDK was randomly chosen from the four directions. Each participant performed two functional localizer runs, and each run comprised 100 trials, resulting in a total of 50 trials per motion direction. The localizer data were used to train motion direction classifiers. This phase took approximately 10 min.

In the exposure phase, we showed participants the four RDKs in either clockwise or counterclockwise order (e.g. 0° → 90° → 180° → 270°, *Figure 1A and C*); each was displayed for 400 ms, followed by 300 ms of a blank screen with a fixation only. Therefore, the full sequence lasted for 2.8 s. The order was counterbalanced among participants, but once decided, it was fixed for each participant. Participants were instructed to detect an oddball RDK by pressing a button, i.e., in 20% trials, dots in one of the four RDKs traveled at a faster speed (9°/s). Each participant completed four runs of 50 trials.

In the main phase, participants were presented with trials in three different conditions: full sequence condition (50% trials), start-only condition (25% trials), and end-only condition (25% trials). Trials in the full sequence condition were identical to those in the exposure phase, which served as 'topping-up' exposure to maintain the exposure effect, similar to 'topping-up' adaptation in visual adaptation studies (*Fang et al., 2005*). In the start- and end-only conditions; however, we only presented the first RDK (start-only condition) or the last RDK (end-only condition) of the full sequence. For a given run, the order of the three conditions was pseudorandomized with the restriction that the start- and end-only trials were always preceded or followed by a full sequence trial. Participants performed an identical oddball detection task as during the exposure phase. The oddball occurred only in 10% of full sequence trials. Finally, each participant completed four runs of 48 trials, yielding a total of 96 full sequence trials, 48 start-only trials, and 48 end-only trials.

### Experiments 2 and 3

In Experiment 2, we only presented the second or third RDK as a cue at the start of the trial, referred to as the second-only and third-only conditions, respectively. The procedure of Experiment 2 was similar to that of Experiment 1, except that the start- and end-only conditions were replaced with the second-only and third-only conditions. Experiment 3 followed the same procedure as Experiment 1, except that the exposure phase was removed.

## Quantification and statistical analysis

### MEG acquisition and preprocessing

Neuromagnetic signals were recorded continuously at 1000 Hz with a 306-channel (204 planar gradiometers; 102 magnetometers), whole-head MEG system (Elekta Neuromag TRIUX) in a magnetically shielded room. Before scanning, four-headed position indicator coils attached to the scalp determined the head position with respect to the sensor array. Coil location was digitized with respect to three anatomical landmarks (nasion and preauricular points) with a 3D digitizer (Polhemus Isotrak system). Participants sat upright inside the scanner, while the stimuli were projected onto a screen suspended in front of them. Participants responded using a MEG-compatible button box held in their right hand.

To reduce noise from external environment, the temporal extension of signal-space separation method was applied at the preprocessing stage using the Elekta Neuromag MaxFilter software

(*Taulu and Simola, 2006*). MEG signals were high-pass filtered at 0.5 Hz using a first-order IIR filter to remove slow drifts. Data were then downsampled from 1000 Hz to 250 Hz for sequenceness analysis or 500 Hz for time-frequency analysis. Excessively noisy segments and sensors were automatically removed before independent component analysis (FastICA, http://research.ics.aalto.fi/ica/fastica) and performed to remove artifacts including cardiac signals, eye movements and blinks, and environmental noise. Artifact components were removed by visual inspection of spatial topography, time course, kurtosis of the time course, and frequency spectrum of all components. Only 72 sensors (including both gradiometers and magnetometers) covering the occipital lobe, labeled as 'occipital' in the MEG data acquisition system, were used for MEG data analyses, except for source localization. The sensor selection was primarily motivated by the main objective of the study, examining replay events in visual cortex.

## Event-related fields

To calculate the ERFs, MEG epochs were segmented around trial onset for each trial and baseline-corrected using the mean activity in the time window of [−0.3 s, 0] before trial onset. Only 48 planar gradiometers of 'occipital' sensors were used to calculate the ERFs. The planar-combined ERF activity was then averaged for each condition. To minimize potential baseline confounds caused by the short interval between every two RDKs, we further calculated the ERF peak amplitude using the mean activity in the time window of [−0.3 s, 0] before the onset of the corresponding RDK as the baseline. Subsequently, peak amplitude during each of the four RDK intervals was calculated as the difference between the peak and its corresponding baseline.

## Multivariate pattern analyses

Multivariate pattern analyses were performed to classify the neural activity patterns elicited by the motion directions of the four RDKs in the main phase. A one-versus-rest Lasso-regularized logistic regression model was trained using the MEG signals from the 72 occipital sensors in the functional localizer phase. Specifically, we trained a five-class classifier, including four classes from trials in which the four RDKs were presented, and an additional class comprising an equivalent amount of null data extracted from the 1–1.5 s ITI. Null data were included to reduce the spatial correlation among the classes, thereby concurrently lowering the decoding probabilities for all classes (*Liu et al., 2021a*; *Nour et al., 2021*). Class weights were balanced by adjusting inversely proportional to class frequencies (i.e. trial numbers) in the training procedure. To reduce noise, MEG signals were averaged across every five trials within the same class before decoding.

The trained classifier was then applied to MEG signals at each time point in the main phase, and decoding probabilities were computed as the outputs of the classifier. Different from the conventional decoding accuracy, where the classifier predicts a single label at each time point, decoding probabilities provide a likelihood estimate for each class. In our study, the classifier outputs a five-column matrix for all trials, where each row represents a single trial and each column represents the probability for one class (i.e. one of the four RDKs and blank ITI). The probabilities in each row sum to 1, reflecting the relative likelihoods across all classes for that trial. The highest probability determines the decoded label when computing decoding accuracy. Finally, averaging these probabilities across trials yields five values that indicate the overall likelihood of the predicted stimulus belonging to a given class.

## Optimal time point of motion direction representation

The optimal time point of motion direction representation was considered as the time point with the highest decoding accuracy in the functional localizer data. To index the optimal time point of each motion direction for each participant, we conducted a time-resolved motion direction decoding analysis on the functional localizer data using Lasso-regularized logistic regression models. A leave-one-trial-out cross-validation procedure was used to train the classifier to determine one of the four motion directions, yielding a decoding accuracy at each time point for each participant. Finally, the time point with the highest decoding accuracy was independently extracted for each participant and each motion direction. These time points are referred to as optimal time points and were used for training feature-specific decoding models.

## Sequenceness measure

To identify sequenceness during the post-cue blank period in each trial, we trained models to detect transient spontaneous neural reactivation of each motion direction representation. Therefore, a one-versus-rest Lasso-regularized logistic regression model was trained separately for each participant and motion direction using the functional localizer data. MEG signals from the 72 occipital sensors obtained throughout all localizer scanning sessions were used to train the decoding models. As our aim was to quantify the evidence of feature-specific sequence, for each motion direction, we trained a binomial classifier, using positive instances from trials in which that feature (e.g. 0°) was presented and negative instances from trials in which all other features (e.g. 90°, 180°, and 270°) were presented, together with an equivalent amount of null data from the 1–1.5 s ITI. The sensor distributions of beta estimates and the spatial correlation among the classifiers are shown in *Figure 3—figure supplement 1*.

The analysis pipeline of sequenceness is illustrated in *Figure 3*. We first applied the trained models to MEG signals at each time point during the blank period to generate a [time × motion direction] reactivation probability (i.e. decoding probability) matrix for each trial (*Figure 3—figure supplement 2*). The TDLM framework was then used to quantify evidence for sequential reactivations consistent with the exposed motion sequence (*Liu et al., 2021b*; *Nour et al., 2021*).

TDLM is a multiple linear regression approach to quantify the extent to which a lagged reactivation time course of one motion direction i (denoted as $X(\Delta t)_i$, where $\Delta t$ indicates lag time) can predict the reactivation time course of another motion direction j (denoted as $X_j$). Two steps were included in this pipeline. First, we performed multiple separate regressions using the reactivation time course of each motion direction j (where $j \in [1:4]$) as the dependent variable. The predictors were the time-lagged reactivation time courses of all four motion directions i (where $i \in [1:4]$). The regression model can be expressed as:

$$X_j = \sum_{i=1}^{4} X\left(\Delta t\right)_i * \beta\left(\Delta t\right)_{i,j} + C. \tag{1}$$

The predictor $X(\Delta t)_i$ was a $\Delta t$-lagged copy of the reactivation time course of $X_i$. The regression coefficients $\beta(\Delta t)_{i,j}$ quantified the strength of the empirical reactivation pattern from motion direction i to motion direction j at a given time lag, $\Delta t$. For example, if $X_j$ represents the reactivation time course of 0° during the blank period in the main phase, and $X_i$ represents the reactivation time course of 90° during the same period, then $\beta(\Delta t)_{i,j}$ is the coefficient that captures the unique variance in the reactivation time course of 0° explained by the $\Delta t$-lagged reactivation time course of 90° (*Figure 3D*, first-level GLM analysis). Finally, all such first-level coefficients were placed in a lag-specific [4×4] empirical transition matrix $\mathrm{B}$, representing evidence for all possible transitions at a specific lag.

In the second step, we quantified the extent to which this empirical transition matrix $\mathrm{B}$ could be predicted by a model transition matrix reflecting the sequence of interest, e.g., the exposed motion sequence (*Figure 3D*, second-level GLM analysis; 1 for transitions of interest and 0 otherwise). We separately modeled forward transitions ($T_F$), which followed the order of the motion sequence, and backward transitions ($T_B$), which followed the reverse order. The strength of these sequences was measured by:

$$\sum_{r=1}^{4} Z_r * T_r. \tag{2}$$

where $\mathrm{B}$ is the [4×4] empirical (lag-specific) transition matrix obtained from the data, $T_r$ is a [4×4] model transition matrix (for regressor r), and $Z_r$ is the scalar regression coefficient quantifying the extent to which the model transition matrix $T_r$ predicts the empirical transitions, $\mathrm{B}$. We included four model transition matrices as regressors: (1) $T_F$: transitions as in the motion sequence in the forward direction (transitions corresponding to [0° → 90° → 180° → 270°]), (2) $T_B$: transitions opposite to the motion sequence in the backward direction ([270° → 180° → 90° → 0°], $T_B$ is the transpose of $T_F$), (3) $T_{auto}$: self-transitions to control for autocorrelation ([4×4] identity matrix), and (4) $T_{const}$: a constant matrix to model away the average of all transitions, ensuring that any weight on $T_F$ and $T_B$ was not due to general background neural dynamics.

Notably, the estimate of sequence strength, $Z_r$, is a relative measure. For instance, a $Z_r$ value of zero for the transition from 0° to 90° does not indicate an absence of replay for that transition; rather, it reflects that the strength of replay of 0° → 90° is not stronger than that of other transitions. Repeating the regression in Equation 2 at each time lag (Δt=4, 8, 12,…, 600 ms) results in both forward (i.e. $Z_1$) and backward (i.e. $Z_2$) sequence strength as a function of time lag. Shorter lags indicate greater time compression, corresponding to faster speeds.

In the current study, sequenceness was defined as the contrast between the evidence for replay of the motion sequence in the forward direction ([0° → 90° → 180° → 270°]) versus the backward direction ([270° → 180° → 90° → 0°]). Specifically, sequenceness was calculated as the difference between the regression coefficients for forward and backward transitions (i.e. $Z_1 - Z_2$). This contrast controls for between-participant variance in the sequential replay per se, which may arise from factors such as task engagement or measurement sensitivity (*Liu et al., 2021a*; *Nour et al., 2021*). As sequenceness is derived from regression coefficients, it is inherently a unitless measure. Positive sequenceness values indicate replay in a predominantly forward direction, whereas negative sequenceness values indicate replay in a predominantly backward direction.

For statistical inference, we used nonparametric permutation tests involving all possible permutations of the stimulus labels at the second-level regression, equivalent to permuting the rows and columns together of the transition matrices used to calculate sequenceness. For each permutation, we calculated the peak absolute mean sequence strength over participants and across lags (controlling for multiple comparisons across lags). Sequence strength in unpermuted data was considered significant if its absolute magnitude was >95% of the within-permutation peak.

## Identifying replay onsets and analyzing time-frequency

Having identified that the replay transition at the group level was a 32 ms-lag, we next identified replay onsets. Replay onset was defined as the time point when a strong reactivation of one motion direction (e.g. 0°) was followed by a strong reactivation of the next motion direction (e.g. 90°) in the sequence, with a 32 ms-lag (*Liu et al., 2019*; *Nour et al., 2021*). We first generated a matrix Orig as

$$Orig = X * T \tag{3}$$

where X is the [time × motion direction] reactivation matrix, and T is the backward transition matrix. The transition matrix T defines the mapping between the motion direction corresponding to column i in X and column i in Orig. For example, column 1 in X is the reactivation time course of the motion direction 0°, while column 1 in Orig is the reactivation time course of the motion direction 90°. We then shifted each column of X by Δt=32 ms to generate another matrix Proj,

$$Proj = X \left( \Delta t \right) \tag{4}$$

where row i in Proj corresponds to row i+32 ms in X. Next, we multiplied Proj and Orig elementwise, summing over the columns of the resulting matrix; therefore, creating a [time × 1] vector, R, in which each element (i.e. row) indicates the strength of replay with a 32 ms-lag at a given time.

$$R_t = \sum_{i=1}^{4} Orig_{ti} * Proj_{ti}. \tag{5}$$

Finally, we identified putative replay event onsets by thresholding R at its 95th percentile, preceded by a 100 ms pre-onset baseline exhibiting a low probability of replay at each time point.

We then epoched MEG data in the blank period surrounding each onset and computed a frequency decomposition (wavelet transformation) in the time window of −100 ms to 150 ms using all sensors. We aimed to find evidence for the power increase in the high-frequency (120–160 Hz) region of interest at replay onset, compared with a pre-onset baseline (−100 ms to −50 ms from onset), similar to a previous study (*Liu et al., 2019*). Finally, we generated two separate [time × frequency] matrices (i.e. using forward and backward transition matrices separately) by averaging estimates over sensors and events, capturing the typical spectrum-resolved power change at replay onset.

## MEG source reconstruction

We identified the neural sources associated with increased ripple power at putative replay onsets. Forward models were generated based on a single shell using the superposition of basic functions that approximately correspond to the plane tangential to the MEG sensor array. LCMV beamforming (*Van Veen et al., 1997*) was used to reconstruct the epoched MEG data to a grid in MNI space (grid step, 5 mm). The sensor covariance matrix for beamforming was estimated using broadband power data across all frequencies. The baseline activity was the mean activity averaged over −100 ms to −50 ms relative to replay onset. All nonartifactual replay epochs were baseline-corrected at source level. We obtained whole-brain results for voxels predicted by participant-specific ripple power at replay onset. Nonparametric permutation tests were performed on the volume of interest to compute the multiple comparison p values of clusters >10 voxels (whole-brain corrected, cluster-defining threshold; t=3.1, n=5000 permutations).

## Statistical analysis

Statistical analyses for MEG data are described in the corresponding Materials and methods section. Specifically, all statistical tests were performed using nonparametric permutation methods. For ERFs, nonparametric cluster-based one-sample t-tests were conducted with a cluster-defining threshold of t=3.1. Clusters spanning more than 10 consecutive time points were considered significant, based on 5000 permutations. For TDLM, nonparametric permutation tests were performed by permuting all possible stimulus label assignments at the second-level regression. Sequenceness was considered significant if its absolute magnitude was >95% of the within-permutation peak. For time-frequency analyses, a nonparametric cluster-based permutation test was applied (cluster-forming threshold: t>3.1; 5000 permutations). For MEG source reconstruction, nonparametric permutation tests were conducted within the volume of interest to identify significant clusters (>10 voxels) using a cluster-defining threshold of t=3.1 and 5000 permutations. All statistical analyses were performed using custom Python scripts.

## Acknowledgements

This study was supported by the National Science and Technology Innovation 2030 Major Program (2022ZD0204802) to FF, the National Natural Science Foundation of China (T2421004, 31930053) to FF, the National Natural Science Foundation of China (32400874) to TH, Beijing Natural Science Foundation (5244044) to TH, and the Young Scientists Fund of the Humanities and Social Science Foundation of Ministry of Education of China (23YJCZH071) to TH.

## Additional information

### Funding

| Funder | Grant reference number | Author |
|---|---|---|
| National Science and Technology Innovation 2030 Major Project | 2022ZD0204802 | Fang Fang |
| National Natural Science Foundation of China | T2421004 | Fang Fang |
| Beijing Natural Science Foundation | 5244044 | Tao He |
| Humanities and Social Science Foundation of Ministry of Education of China | 23YJCZH071 | Tao He |
| National Natural Science Foundation of China | 32400874 | Tao He |

| Funder | Grant reference number | Author |
|---|---|---|
| National Natural Science Foundation of China | 31930053 | Fang Fang |

The funders had no role in study design, data collection and interpretation, or the decision to submit the work for publication.

## Author contributions

Tao He, Conceptualization, Data curation, Formal analysis, Funding acquisition, Investigation, Visualization, Writing - original draft, Writing - review and editing; Xizi Gong, Data curation, Validation, Investigation, Methodology, Writing - review and editing; Qian Wang, Data curation, Supervision, Investigation, Writing - review and editing; Xinyi Zhu, Data curation, Investigation; Yunzhe Liu, Supervision, Investigation, Methodology, Writing - review and editing; Fang Fang, Resources, Supervision, Funding acquisition, Project administration, Writing - review and editing

## Author ORCIDs

Tao He ⓘ http://orcid.org/0000-0002-1009-0500
Xizi Gong ⓘ http://orcid.org/0009-0005-6263-0774
Qian Wang ⓘ http://orcid.org/0000-0003-2347-8798
Xinyi Zhu ⓘ http://orcid.org/0000-0001-7722-5150
Yunzhe Liu ⓘ http://orcid.org/0000-0003-0836-9403
Fang Fang ⓘ https://orcid.org/0000-0002-7718-2354

## Ethics

The experiments reported here were carried out in accordance with the guidelines expressed in the Declaration of Helsinki. All participants provided written informed consent in accordance with the procedures and protocols approved by the Human Subject Review Committee of Peking University (#2021-10-13).

Reviewer #1 (Public review): https://doi.org/10.7554/eLife.101511.4.sa1
Reviewer #2 (Public review): https://doi.org/10.7554/eLife.101511.4.sa2
Author response https://doi.org/10.7554/eLife.101511.4.sa3

# Additional files

## Supplementary files
MDAR checklist

## Data availability

TDLM MATLAB code is available on GitHub (copy archived at *Liu, 2021*). Custom code and data have been deposited at the Open Science Framework (https://osf.io/hdjtr/).

The following dataset was generated:

| Author(s) | Year | Dataset title | Dataset URL | Database and Identifier |
|---|---|---|---|---|
| He T, Gong X, Wang Q, Zhu X, Liu Y, Fang F | 2025 | Non-feature-specific elevated responses and feature-specific backward replay in human brain induced by visual sequence exposure | http://doi.org/10.17605/OSF.IO/HDJTR | Open Science Framework, 10.17605/OSF.IO/HDJTR |

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
