## [Editor Report · eLife Assessment]

This **valuable** study investigates both online responses to, and offline replay of, visual motion sequences. Sophisticated MEG analyses provide **convincing** evidence for both feature-specific and non-specific sequence representations. These intriguing findings will be of interest to perception and learning researchers alike.

---

## [Referee Report · Reviewer #1 (Public review)]

Summary:

The study identifies two types of activation: one that is cue-triggered and non-specific to motion directions, and another that is specific to the exposed motion directions but occurs in a reversed manner. The finding that activity in the medial temporal lobe (MTL) preceded that in the visual cortex suggests that the visual cortex may serve as a platform for the manifestation of replay events, which potentially enhance visual sequence learning.

Evaluations:

Identifying the two types of activation after exposure to a sequence of motion directions is very interesting. The experimental design, procedures and analyses are solid. The findings are interesting and novel.

In the original submission, it was not immediately clear to me why the second type of activation was suggested to occur spontaneously. The procedural differences in the analyses that distinguished between the two types of activation need to be a little better clarified. However, this concern has been satisfactorily addressed in the revision.

---

## [Referee Report · Reviewer #2 (Public review)]

This paper shows and analyzes an interesting phenomenon. It shows that when people are exposed to sequences of moving dots (That is moving dots in one direction, followed by another direction etc.), that showing either the starting movement direction, or ending movement direction causes a coarse-grained brain response that is similar to that elicited by the complete sequence of 4 directions. However, they show by decoding the sensor responses that this brain activity actually does not carry information about the actual sequence and the motion directions, at least not on the time scale of the initial sequence. They also show a reverse reply on a highly-compressed time scale, which is elicited during the period of elevated activity, and activated by the first and last elements of the sequence, but not others. Additionally, these replays seem to occur during periods of cortical ripples, similar to what is found in animal studies.

These results are intriguing. They are based on MEG recordings in humans, and finding such replays in humans is novel. Also, this is based on what seems to be sophisticated statistical analysis. The statistical methodology seems valid, but due to its complexity it is not easy to understand. The methods especially those described in figures 3 and 4 should be explained better.

Comments on second revised version by editorial team:

In response to the reviewer, the authors have substantially expanded and clarified their description of the methodology in this version of the manuscript.

---

## [Author Response]

The following is the authors’ response to the previous reviews

**Public Reviews:**

**Reviewer #1 (Public review):**
Summary:The study identifies two types of activation: one that is cue-triggered and nonspecific to motion directions, and another that is specific to the exposed motion directions but occurs in a reversed manner. The finding that activity in the medial temporal lobe (MTL) preceded that in the visual cortex suggests that the visual cortex may serve as a platform for the manifestation of replay events, which potentially enhance visual sequence learning.Evaluations:Identifying the two types of activation after exposure to a sequence of motion directions is very interesting. The experimental design, procedures and analyses are solid. The findings are interesting and novel.In the original submission, it was not immediately clear to me why the second type of activation was suggested to occur spontaneously. The procedural differences in the analyses that distinguished between the two types of activation need to be a little better clarified. However, this concern has been satisfactorily addressed in the revision.

We thank the reviewer for his/her positive evaluation and thoughtful comments.

**Reviewer #2 (Public review):**
This paper shows and analyzes an interesting phenomenon. It shows that when people are exposed to sequences of moving dots (That is moving dots in one direction, followed by another direction etc.), that showing either the starting movement direction, or ending movement direction causes a coarsegrained brain response that is similar to that elicited by the complete sequence of 4 directions. However, they show by decoding the sensor responses that this brain activity actually does not carry information about the actual sequence and the motion directions, at least not on the time scale of the initial sequence. They also show a reverse reply on a highly-compressed time scale, which is elicited during the period of elevated activity, and activated by the first and last elements of the sequence, but not others. Additionally, these replays seem to occur during periods of cortical ripples, similar to what is found in animal studies.These results are intriguing. They are based on MEG recordings in humans, and finding such replays in humans is novel. Also, this is based on what seems to be sophisticated statistical analysis. The statistical methodology seems valid, but due to its complexity it is not easy to understand. The methods especially those described in figures 3 and 4 should be explained better.

We thank the reviewer’s detailed evaluation. As suggested, we have further revised the Methods and Results sections, particularly the descriptions related to Figures 3 and 4, to enhance clarity. Please see the revisions highlighted in red in the revised manuscript.

**Recommendations for the authors:**

**Reviewer #2 (Recommendations for the authors):**
The most important results here are in Figure 4, and they rely on methods explained in Figure 3. Figure 4 and the results in the figure are confusing.What is the red bar in 4B,E. What are the units of the Y axis in figure 4B,E?Does sequenceness have units? How do we interpret these magnitudes apart from the line of statistical significance? Shouldn't there be two lines, one for forward replay and the other for backward replay rather than a single line with positive and negative values? The term sequnceness is defined in figure 3, and is key. The replayed sequence in figure 4A,D seems to last about 120 ms.What is the meaning of having significance only within a window of 28-36 ms?

We thank the reviewer’s careful reading and insightful comments. We apologize for the lack of clarity regarding these details in the previous version. As mentioned above, we have revised the Methods and Results sections to enhance clarity throughout the manuscript. For convenience, we provide detailed explanations addressing the specific points raised by the reviewer below.

First, the red bars in Figures 4B and 4E indicate the lags when the evidence of sequenceness surpassed the statistical significance threshold, as determined by permutation testing. We have now explicitly clarified this in the revised figure captions.

Second, sequenceness doesn’t have units. It corresponds to the regression coefficient (β) obtained from the second-level GLM in the TDLM framework. Specifically, in the first step of TDLM, we constructed an empirical transition matrix that quantifies the evidence for all possible transitions (e.g., 0° → 90°) at each time lag (Δt). In the second step, we evaluated the extent to which each model transition matrix (e.g., forward or backward transitions) predicts the empirical transition matrix at each Δt, yielding second-level β values. Sequenceness is defined as the difference between the β values for the forward and backward transition models, reflecting the relative strength and directionality of sequential replay. As it is derived from regression coefficients, sequenceness is inherently a unitless measure.

Regarding the interpretation of sequenceness magnitudes beyond statistical significance, the β values reflect the extent to which the model transition matrix explains variance in the empirical transition matrix. While larger β values suggest stronger sequenceness, absolute magnitudes are influenced by various factors, such as between-participant noise. Therefore, the key criterion for interpreting these values is whether they surpass permutationbased significance thresholds, which indicate that the observed sequenceness is unlikely to have occurred by chance.

Third, as the reviewer correctly pointed out, we initially computed two separate regression lines, one for forward replay and the other for backward replay. We then defined sequenceness as the contrast between the forward and backward replay (forward minus backward). This contrast approach is commonly used in previous studies to remove between-participant variance in the sequential replay per se, which may arise due to variability in task engagement or measurement sensitivity (Liu et al., 2021; Nour et al., 2021).

Finally, regarding the duration of replay events, the example sequences shown in Figures 4A and 4D indeed span about 120 ms in total. However, the time lag (Δt) between successive reactivation peaks within these sequences is about 30 ms. This is in line with the findings shown in Figures 4B and 4E, where statistical significance is observed at a time lag window of 28 – 36 ms on the x-axis. It is important to note that the x-axis in these plots represents the time lag (Δt) between sequential reactivations, rather than absolute time.

We hope these clarifications address the reviewer’s concerns, and we have revised the manuscript accordingly to make these points clearer to readers.

The methods here are not simple and not simple to explain. The new version is easier to understand. From the new version it seems that the methodology is sound. It should be still clarified and better explained.

We have carefully revised the manuscript to better explain the methodology. We appreciate the reviewer’s feedback, which is valuable in improving the clarity of our work.

Now that I understand what they mean by decoding probability, I think that this term is confusing or even misleading. The decoding accuracy is the probability that the direction of motion classification was correct. It seems the so-called decoding probability is value of the logistic regression after normalizing the sum to 1. If this is a standard term it can probably be kept, if not another term would be better.

Thank you for the reviewer’s comment. We agree that the term decoding probability may initially seem confusing. However, decoding probability is a commonly used term in the neural decoding literature, particularly in human studies (e.g., Liu et al., 2019; Nour et al., 2021; Turner et al., 2023). To maintain consistency with previous work, we have kept this term in the manuscript. We appreciate the opportunity to clarify this point.

References

Liu, Y., Dolan, R. J., Higgins, C., Penagos, H., Woolrich, M. W., Ólafsdóttir, H. F., Barry, C., Kurth-Nelson, Z., & Behrens, T. E. (2021). Temporally delayed linear modelling (TDLM) measures replay in both animals and humans. eLife, 10, e66917. https://doi.org/10.7554/eLife.66917

Liu, Y., Dolan, R. J., Kurth-Nelson, Z., & Behrens, T. E. J. (2019). Human Replay Spontaneously Reorganizes Experience. Cell, 178(3), 640-652.e14. https://doi.org/10.1016/j.cell.2019.06.012

Nour, M. M., Liu, Y., Arumuham, A., Kurth-Nelson, Z., & Dolan, R. J. (2021). Impaired neural replay of inferred relationships in schizophrenia. Cell, 184(16), 4315-4328.e17. https://doi.org/10.1016/j.cell.2021.06.012

Turner, W., Blom, T., & Hogendoorn, H. (2023). Visual Information Is Predictively Encoded in Occipital Alpha/Low-Beta Oscillations. Journal of Neuroscience, 43(30), 5537–5545. https://doi.org/10.1523/JNEUROSCI.0135-23.2023